# Real-Time Monitoring of the Quality Changes in Shrimp (*Penaeus vannamei*) with Hyperspectral Imaging Technology during Hot Air Drying

**DOI:** 10.3390/foods11203179

**Published:** 2022-10-12

**Authors:** Wenya Xu, Fan Zhang, Jiarong Wang, Qianyun Ma, Jianfeng Sun, Yiwei Tang, Jie Wang, Wenxiu Wang

**Affiliations:** College of Food Science and Technology, Hebei Agricultural University, Baoding 071000, China

**Keywords:** shrimp, hot air drying, quality change, hyperspectral images, low field magnetic resonance

## Abstract

Hot air drying is the most common processing method to extend shrimp’s shelf life. Real-time monitoring of moisture content, color, and texture during the drying process is important to ensure product quality. In this study, hyperspectral imaging technology was employed to acquire images of 104 shrimp samples at different drying levels. The water distribution and migration were monitored by low field magnetic resonance and the correlation between water distribution and other quality indicators were determined by Pearson correlation analysis. Then, spectra were extracted and competitive adaptive reweighting sampling was used to optimize characteristic variables. The grey-scale co-occurrence matrix and color moments were used to extract the textural and color information from the images. Subsequently, partial least squares regression and least squares support vector machine (LSSVM) models were established based on full-band spectra, characteristic spectra, image information, and fused information. For moisture, the LSSVM model based on full-band spectra performed the best, with residual predictive deviation (RPD) of 2.814. For *L**, *a**, *b**, hardness, and elasticity, the optimal models were established by LSSVM based on fused information, with RPD of 3.292, 2.753, 3.211, 2.807, and 2.842. The study provided an in situ and real-time alternative to monitor quality changes of dried shrimps.

## 1. Introduction

Shrimp (*Penaeus vannamei*) harvesting is one of the most economically significant fishing activities in China attracting attention from consumers due to the high protein content and rich nutritional composition of shrimp [1,2]. According to the China Fisheries Statistical Yearbook, the *Penaeus vannamei* aquaculture production in China was 1.1977 millions of tons in 2020. However, the shrimp harvest suffers from rapid deterioration due to biochemical reactions and microbial activity after death [3,4,5], which directly affect its shelf life. Hot air drying, a common and practical method of drying seafood, can prolong the shelf life of the shrimp harvest [6,7,8]. As a foodstuff, dried shrimp has the advantages of a unique flavor, rich nutrition, easy storage, and high consumer demand [9,10]. However, drying is a complex process involving water evaporation, protein degradation and denaturation, and the formation of flavor compounds [11,12]. Ineffective drying can adversely impact the color, texture, and nutrition attributes of the dried shrimp product [13]. Therefore, it is imperative to monitor and control critical quality parameters during the drying process to ensure consistency among batches, as well as uniformity of the end-product.

Current analytical methods employed to measure these quality characteristics in factories, such as oven drying and texture profile analysis (TPA), are time-consuming, destructive, cumbersome, and restricted to off-line usage [14,15,16]. Therefore, it is necessary to develop an effective, rapid, nondestructive, and real-time detection method for dried shrimp quality control. With the development of optical and spectroscopic technologies, hyperspectral imaging (HSI) has been successfully applied to evaluate food safety and quality, monitor food preparation processes, and identify adulteration [17,18]. HSI simultaneously captures both spectral and spatial information of a sample by integrating spectroscopic and computer vision or imaging techniques into one system [19,20,21]. Another unique characteristic is that HSI generates a visual distribution map of reference values to enable the prediction and quantification of internal sample constituents, as well as the simultaneous determination of their location on the sample surface [22,23].

Based on these advantages, HSI has been applied to monitor quality changes in meat, fruit, vegetable, and cereal foods during drying. For example, Sun et al. used HSI to monitor the moisture contents of scallops during drying, and reported a model prediction accuracy of greater than 0.9 [24]. Moreover, Netto et al. used HSI to evaluate the water uniformity of the melon drying process under different pretreatments by visualizing the moisture content in the samples [25]. However, most existing studies only employ spectral information for quality indicator evaluation, ignoring image information, such as color and texture in their modeling. To improve prediction accuracy, the importance of combining spectral and spatial HIS information has been emphasized by several researchers. This technique has been used to discriminate between different breeds of chicken [26], predict the storage time and moisture content of cooked beef [19], and assess the fat and moisture contents of salmon [27]. The results indicate that a combination of spectral and spatial HSI data is more comprehensive and intuitional than conventional analyses. Furthermore, considering that the shrimp drying process involves color and texture changes, it is crucial to include image information in the spectral model for quality control. To the best of our knowledge, there are no previous data fusion studies on the visualization of moisture and other quality indicators in dried shrimp. Additionally, previous studies only predicted moisture and other quality indicator contents, neglecting the link between moisture distribution and other quality characteristics, which may clarify the mechanisms governing shrimp quality changes during the drying process.

Therefore, the purpose of the current study is to explore the correlation between shrimp water distribution state and other quality indices and combine spatial and spectral information of the hypercube to measure shrimp quality changes during the drying process. The specific objectives are: (1) to quantify changes in shrimp during hot air drying through moisture content measurement, color properties (*L**, *a**, *b**) analysis, and texture profile analysis (hardness, adhesiveness, elasticity, stickiness, and chewiness); (2) to monitor the dynamic water sate and water migration by low field magnetic resonance (LF-NMR) and determine the correlation between water distribution and other quality indicators by Pearson correlation analysis; (3) to acquire hyperspectral reflectance images of shrimps at different drying stages, as well as spectral data and color and textural features from the region of interest (ROI); (4) to establish partial least squares regression (PLSR) and least squares support vector machine (LSSVM) models based on spectral, image, and fusion information; and (5) to visualize shrimp quality at the pixel level using the optimal models.

## 2. Materials and Methods

### 2.1. Sample Preparation and Drying Experiments

Live shrimp (*Penaeus vannamei)*, each weighing approximately 15 ± 3 g, were purchased from a local market in Baoding, China. The live shrimp were transported to the laboratory within 30 min and stored in ice water, until their death. The shrimp were boiled in salt water with a mass fraction of 3% for 2 min and removed to dry the surface moisture. Then, they were used for constant drying experiments at 55 °C using an electrical blast drying oven. When the moisture content had been reduced by approximately 35%, the taste of the dried shrimp was optimal. The total dry processing time was 12 h, and shrimp sample were collected after boiling and drying for 1, 2, 3, 4, 5, 6, 7, 8, 9, 10, 11, and 12 h. A total of 16 samples were collected at each sampling point, half were used for color, moisture determination, and hyperspectral measurement, and the other half were used for texture analysis, LF-NMR analysis, magnetic resonance imaging (MRI) measurement, and hyperspectral measurement. Thus, 104 samples (8 samples × 13 drying levels) were involved for each quality indicator prediction.

### 2.2. Quality Indicators Determination

#### 2.2.1. Moisture Content Measurement

The direct drying method found in the *National Standards of China (GB 5009.3-2016)* was used to calculate the moisture content of the shrimp samples. The specific steps employed were as follows. First, the glass flat weighing bottles were placed in a 105 °C oven to dry to a constant weight. Second, 3.0 g of shrimp samples for each drying period were placed into weighing bottles and the total mass of the bottles and shrimp samples were accurately weighed, noted as *m*_1_. Third, the bottles were placed in an oven at 105 °C for drying and removed after 1 h, then weighed after cooling in the desiccator for half an hour. These steps were repeated until the total mass was not changing and the final mass was weighed accurately and noted as *m*_2_. The moisture content of the shrimp samples at each time point during the drying process was calculated as follows:(1)$$X=\frac{m_{1}{-m}_{2}}{m_{1}{-m}_{3}}\times 100\%$$
where *X* (unit: g/100 g) indicates the moisture content of the shrimp samples at each time point during the drying process; *m*_1_ and *m*_2_ (unit: g) indicate the mass of the weight bottles and samples before and after drying, respectively; and *m*_3_ (unit: g) indicates the mass of the weight bottles.

#### 2.2.2. Color Analysis

The color of the shrimp samples was evaluated using a CR-400 color difference meter (Konica Minolta Co., Ltd., Tokyo, Japan) after equilibration to room temperature. The second abdominal segment of the shrimp was used for color measurement. The color differences were analyzed using lightness (*L**), green to red (*a**), and blue to yellow (*b**). All experiments were conducted eight times.

#### 2.2.3. Texture Profile Analysis (TPA)

The shrimp with the head and shell removed were subjected to texture analysis. Texture variables, including hardness, elasticity, stickiness, adhesiveness, and chewiness, were obtained using Texture Expert software (TMS-Pro, Food Technology Corporation, Sterling, VA, USA) The measurement parameters were set to TPA mode; the probe type was P/5, compression ratio was 45%, detection rate was 30 mm/min, shape variable was 60%, minimum force was 0.5 N, and return distance was 2.5 cm. Each shrimp sample was measured eight times at each point during the drying processes.

### 2.3. LF-NMR Transverse Relaxation Measurements

The relaxation measurements were performed on a Meson NMI20-040H-I LF-NMR analyzer (NMI20-040H-I, NIUMAG Electronic Technology Co., Ltd., Shanghai, China) with a magnetic field strength of 0.5 T and corresponding resonance frequency for protons of 20 MHz. The shrimp samples were placed in a cylindrical glass tube, and a 30-mm diameter radio frequency coil was used to collect Carr–Purcell–Meiboom–Gill sequence (CPMG) decay signals, with a π-value (the time between pulses 90 and 180) of 200 μs; the lengths of these two pulses were 9.52 μs and 18.48 μs, respectively. The repetition time between two scans was 1500 ms. Distributed multiexponential fitting analysis was performed on the T_2_ relaxation data using MultiExp Inv Analysis software (NIUMAG Electronic Technology Co., Ltd., Shanghai, China). The T_2_ relaxation spectra were obtained from this analysis; the lateral and vertical axes represent the relaxation time and signal intensity, respectively (corresponding to the proportion of water molecules exhibited at that relaxation time).

### 2.4. MRI Analysis

MRI was also performed using a Meson NMI20-040H-I LF-NMR analyzer ((NMI20-040H-I, NIUMAG Electronic Technology Co., Ltd., Shanghai, China)) equipped with a 60-mm radio frequency coil. A spin echo (SE) sequence was applied to obtain T_2_ weighted images of the shrimp. The following scanning parameters were used: field of view (FOV) = 100 mm × 100 mm, slice width = 1.1 mm, slice gap = 1.1 mm, average = 8, read size = 256, phase size = 192, T_2_ weighted image echo time (TE) = 20 ms, and repetition time (TR) = 500 ms.

### 2.5. Hyperspectral Image Acquisition and ROI Selection

Images of shrimp were acquired using a pushbroom HSI system in the reflectance mode. The system consisted of 4 components: a charge-coupled device (CCD) camera (FX 10, Specim Ltd., Helsinki, Finland) with a resolution of 1024 pixels in the spatial dimension and 224 bands in the spectral dimension, hyperspectral imaging workstation with a spectral range of 400–1000 nm, 2 halogen lamps, and computer with hyperspectral image analysis software. The spectral resolution was 5.5 nm, and the imaging speed of full band acquisition was 330 Frames Per Second (FPS). Before the experiment, the instrument was preheated for 30 min to ensure its stability. The samples were placed on a mobile platform for image acquisition. To prevent image oversaturation, it was necessary to set the speed of the moving platform, camera exposure time, and acquisition distance in advance; after repeated testing, these three parameters were set to 7.5 mm/s, 50 ms, and 30 cm, respectively. Simultaneously, black and white correction of the acquired hyperspectral image was conducted to reduce the influence of the dark current of the CCD camera and uneven brightness of the light source. The correction formula is given by:(2)$$R=\frac{R_{o}-R_{d}}{R_{w}-R_{d}}$$
where *R*_o_ represents the original spectral image, *R*_w_ represents the whiteboard image, and *R*_d_ represents the darkfield image.

ROI spectral extraction of the hyperspectral image was performed using ENVI 5.2 software (Exelis Visual Information Solutions Co., Boulder, CO, USA). All pixels, except those corresponding to the shrimp head and tail, were selected to contain as much information as possible about the sample. As the collected spectral images were clear in all bands, the entire spectral range with 224 bands was retained for analysis.

### 2.6. Spectral Pre-Processing and Optimal Wavelengths Selection

Spectral preprocessing involves the use of appropriate mathematical analyses to correct random noise in the spectra and light scattering generated by the instruments, which is helpful for highlighting valuable spectral information [28]. In current work, Savitzky–Golay smoothing and standard normal variable transformation (SNV) method were employed to remove the interference information from the spectra. Meanwhile, among the collected spectral information, the spectral data of certain bands could be explained or replaced by those of other bands. This situation caused a large amount of redundant information in the spectrum. Owing to the existence of redundant information, the prediction accuracy of the established model decreased; as the computational burden increased, the computational speed decreased. To overcome these problems, it was important to select a small set of optimal wavelengths that reflected the changes in quality to establish the model. The competitive adaptive reweighting sampling (CARS) method was used to select the optimal wavelengths in this study.

### 2.7. Image Color and Texture Information Extraction

Compared with traditional spectroscopic methods, HSI has the advantage of providing abundant image information related not only to size and shape, but also color and textural features. Color moments represent a simple and effective means of representing the color features, with the first, second, and third order moments commonly used to express the color distribution of images. Because of its advantages of low feature vector dimensionality and no color space quantization, color moments are effective for characterizing color distributions in images [29]. In this study, we extracted the *RGB* (red, green, and blue) images synthesized from the hyperspectral images at 647 nm, 550 nm and 460 nm using ENVI 5.2 (Exelis Visual Information Solutions Co., Boulder, CO, USA), and the first-order moment and second-order moment information were calculated. Then, the *RGB* tricolor was transformed into HSV (hue, saturation, and value) mode, which is based on the intuitive properties of color, to extract three additional feature variables. Finally, nine color features were obtained to reflect the image difference of samples with different drying levels.

The gray-level co-occurrence matrix (GLCM) method was used to extract the texture information about the shrimp images. Four special mutually independent features of contrast, correlation, energy, and homogeneity were used to describe the co-occurrence matrix data in four orientations of 0°, 45°, 90°, and 135°, and the distance of each pixel pair was set to 1. The contrast value expresses local variations in the gray levels of the GLCM, the correlation measures the image linearity among pixels, the homogeneity measures the density of the distribution of elements in the GLCM to its diagonal, and the energy measures the textural uniformity of the image [30]. All textural values based on the different directions were then averaged into one value representing the textural features of the sample for subsequent analysis. Before constructing the texture matrix, principal component analysis (PCA) was performed to select the optimal characteristic images [31]. The implementation procedures for PC images were performed using the ENVI 5.2 software (Exelis Visual Information Solutions Co., Boulder, Colorado, USA), and the color and texture feature extraction were performed in Matlab 2012a (MathWorks Co., Natick, MA, USA).

### 2.8. Quantitative Analysis Models

In this study, PLSR and LSSVM techniques were compared to establish the quantitative relationships between spectroscopic data and image information and the measured moisture content, *L**, *a**, *b**, hardness, and elasticity during the drying process. The 104 samples were divided 3:1 into calibration and prediction sets for L and LSSVM modeling. PLSR is an effective multivariate regression method that enables regression modeling of multiple independent variables; it is particularly effective when the variables are highly linearly correlated [32]. The LSSVM technique can be applied to both linear and nonlinear regression models. For nonlinear regression problems, the LSSVM approach first performs nonlinear mapping from the input space onto a high-dimensional feature space using a nonlinear kernel function. This method then performs linear regression in the same feature space, which can be used to solve linear regression problems [33]. The predicted results were compared with the actual values, and the model performance was evaluated in terms of the correlation coefficient (*R*), root-mean-squared error of calibration set (RMSEC) and root-mean-squared error of prediction set (RMSEP), and residual predictive deviation (RPD). The afore-mentioned data analyses were implemented using Matlab2012a (MathWorks Co., Natick, MA, USA).

### 2.9. Visualization of Shrimp Quality Indicators

The advantage of HSI is its ability to transfer multivariate spectral data in a pixel-wise manner by inputting the spectra in each pixel into an established calibration model. In this study, we selected the final optimal models of moisture content, *L**, *a**, *b**, hardness, and elasticity for visualization by pseudo-color data processing. All visualization steps were executed in Matlab2012a (MathWorks Co., Natick, MA, USA). The key steps of the analysis procedure are summarized in Figure 1.

### 2.10. Statistical Analysis

The physicochemical data were statistically analyzed using the Statistical Package for the Social Sciences (SPSS) version 18.0 software package (SPSS Inc., Chicago, IL, USA). Data are expressed as mean ± standard deviation (SD), and significance was defined as *p* < 0.05. The correlation between the LF-NMR results and the physicochemical parameters was determined by Pearson correlation analysis.

## 3. Results and Discussion

### 3.1. Quality Indicators Analysis

#### 3.1.1. Moisture Content Analysis

The changes in the moisture content and drying rate of shrimp during the hot air drying process are shown in Figure 2. The moisture content of the fresh shrimp was 75.87%. The drying endpoint is 12 h at which point the moisture content decreases to 35.02%. It can be seen from Figure 2 that within 2 h of drying, the moisture content decreases at a slower rate; the drying rate at this point is 3.74% w.b h^−1^; during 2–8 h, the moisture content decreases at a faster rate and the drying rate reaches a maximum of 5.72% w.b h^−1^ at 4 h; and after 8 h, the drying rate decreases slowly. The reason for this phenomenon could be that, in the early drying stage, due to the high moisture content of shrimp, the oven space was saturated, and the moisture on the surface of the shrimp could not evaporate in time [34]. This situation increased the humidity in the oven. With further hot air drying, the protein denatured because of the heat, which reduced the interaction between matter and water. The release of water and increase in the drying rate could also have been due to fiber shrinkage, leading to decreased intracellular spaces and thus facilitating evaporation. Similar results were obtained by Sun et al., who found that the moisture content of scallops decreased by approximately 50% during drying at 55 °C for 5 h and that the decrease in moisture content was mainly associated with free water migration [24]. When the moisture on the surface of the shrimp evaporated, the free water in the body evaporated to a certain extent, and the remaining bound water could not easily flow and evaporate, resulting in a slow decline in the moisture content and drying rate. Shi et al. found that the decrease in the moisture content of beef jerky with increased drying time and temperature was related to the degree of moisture migration [35]. The current results corroborate these findings.

#### 3.1.2. TPA

Texture analysis of the shrimp was performed during the drying process, and the results are shown in Table 1. The TPA parameters include hardness, adhesiveness, elasticity, stickiness, and chewiness. As shown in Table 1, the hardness of the shrimp samples significantly increased with increasing drying time (*p* < 0.05) due to the change of drying rate and the shrinkage of shrimp muscle fibers during the drying process. Latorre et al. explained that the dissociation of actin and myosin, disintegration of muscle fibers, and myofibril dissociation lead to the formation of small fragments and increase the hardness of the disordered structure of the muscle fiber [36]. The shrimp showed the least adhesiveness after drying for 12 h, which highlighted their improved fragility. The elasticity of shrimp firstly increased and then decreased during drying and reached its highest value at 7 h of drying. The changes in elasticity were related to the contraction of the muscle fiber. As muscle fiber contracts, the muscle proteins form a dense reticular structure, which is prone to irreversible deformation when the muscle tissue is extruded, and the spatial structure of muscle proteins is small, resulting in reduced elasticity [37]. In addition, the stickiness and chewiness of shrimp samples increase with increasing drying time. Chewiness and stickiness are parameters used in comprehensive analysis. Chewiness represents the energy required to chew solid samples, whereas stickiness represents the energy required to separate food from its contact material. As hardness and elasticity showed significant changes during drying, they were chosen as representative indicators of TPA for further modeling.

#### 3.1.3. Color Analysis

The market value of shrimp depends on the visual appearance of their body color, which is attributed to the presence of astaxanthin [38]. This carotenoid pigment is responsible for orange red tissue pigmentation in shrimp meat. Table 2 shows the color differences of the shrimp. The *L** value of shrimp increases from 40.71 when fresh to 63.85 after boiling (*p* < 0.05), which may be due to the increase in heat during boiling, resulting in protein accumulation and an increase in opacity. However, the *L** value of the shrimp decreases with more drying (*p* < 0.05). The blackening of dried shrimp is attributed to the Maillard reaction during drying [39]. Moreover, *a** and *b** exhibit similar trends throughout drying. The *a** and *b** values of dried shrimp are significantly higher than those of fresh shrimp (*p* < 0.05). The formation of redness upon the exposure of shrimp meat to heat is a result of the release of astaxanthin owing to the breakdown of carotene protein during denaturation. There are slight decreases in *a** and *b** values in the late drying period, which may be due to a slower drying rate and longer drying time, resulting in the slight damage of astaxanthin from the extension of hot air-drying. Regarding Δ*E* values, the results for Δ*E* > 12 show that the color of shrimp during drying is notably different from that of fresh shrimp.

### 3.2. LF-NMR Analysis

LF-NMR spectroscopy measures the absorption of radio frequency resonance in presence of an external magnetic field [30]; thus, the spin–spin relaxation time (T_2_) is closely related to the water state and dynamics in foods. Protons of all substances are surrounded by a small magnetic field; thus, each proton creates a tiny magnetic field that is affected by the magnetic field of other protons [40]. Therefore, as the T_2_ of a sample is small or large when the distance between protons is relatively small or large, respectively, T_2_ value analysis is a fast and effective method that allows to identify changes in moisture content and status, and reflects (to some extent) the micro-molecular structure of a sample [41,42]. Since water can alter the interaction between the different components of foods, drying can significantly modify the microstructure of foods. Herein, the T_2_ signal amplitude of shrimp at different drying stages was measured to characterize the change of water state (Figure 3a). To better investigate the water state in the different samples, the relaxation times T_21_, T_22_, and T_23_ of shrimp were defined as bound water that was tightly attached to macromolecules when T_21_ was 0.01–10 ms, immobilized water entrapped within the extra-myofibrillar lattice when T_22_ was 10–100 ms, and free water when T_23_ was 100–10,000 ms, respectively. Noteworthily, the levels of bound water, immobilized water, and free water quickly decreased, as denoted by the shift of the main peaks and signal amplitudes to the left direction with increased drying time. These results indicate that the remaining water molecules within the shrimp samples form strong adsorption connections with the dry matter. The strongest T_2_ signal amplitudes were observed in fresh and boiled shrimp, mainly due to their free and immobilized water, whereas the signals of free water gradually disappeared and those of bound and immobilized water decreased as drying proceeded (Figure 3a). Moreover, the relaxation times of T_21_ and T_22_ decreased from 3.05 to 1.52 and 28.48 to 14.17, respectively, and T_23_ became 0 ms after 9 h of drying (Figure 3b), which indicates that the free water is the main moisture lost during drying. Therefore, the LF-NMR results revealed that the mobility of the bound, immobilized, and free water molecules is reduced due to shrimp muscle contraction and the marked evaporation of free water during the drying process.

### 3.3. MRI Analysis

The hydrogen proton MRI has been used as a noninvasive method to evaluate the distribution of moisture content within food products [43]. The T_2_ weighted images taken at the transverse geometric center of each sample during the drying process revealed the distribution of the water within high-mobility protons (Figure 4). With increasing drying time, a continuous decrease is observed in the size of the brighter regions, suggesting the loss of a longer relaxation signal of water during drying. In addition, a decrease in the signal intensity from the external surface to the inner region is evident. Similar phenomenon was also observed by Ling et al. who found that the red region gradually changed to blue, and the color and size of the blue region remarkably decreased from the exterior to the interior part with increasing drying time of shrimp [39]. These results confirmed that the water relaxation signal gradually weakens and the water content continuously decreases during the drying process, in agreement with the above-described changes observed in moisture content in shrimp during drying.

### 3.4. Correlation between LF-NMR and Physicochemical Properties

As a rapid, noninvasive method, LF-NMR relaxation is often applied to investigate water mobility in materials and foods [44]. However, the correlation between water distribution state and the physicochemical parameters of shrimp during the drying process needs deeper exploration. Therefore, the relationship between T_2_ relaxation times (T_21_, T_22_, and T_23_) and shrimp physicochemical properties was determined by Pearson correlation analysis (Figure 5). The results indicated good correlations between LF-NMR data and shrimp moisture content, hardness, adhesiveness, stickiness, and chewiness. Specifically, moisture content was significantly positively correlated with T_21_ (*R* = 0.943), T_22_ (*R* = 0.914), and T_23_ (*R* = 0.903), which may be explained by the substantial effect of moisture on the proteins and myofibril in shrimps. This was similar to the findings of Cheng et al., who also reported a positive correlation between the decrease in moisture content and the change in relaxation times [42]. Regarding shrimp texture, its hardness, stickiness, and chewiness were negatively correlated with T_21_ (*R* = −0.877, *R* = −0.889, *R* = −0.852) and T_23_ (*R* = −0.846, *R* = −0.875, *R* = −0.844), whereas adhesiveness was positively correlated with T_21_ (*R* = 0.832) and T_23_ (*R* = 0.872). These results agree with those of Wang et al., who reported that T_22_ was highly correlated (*p* < 0.01) with hardness, elasticity, and chewiness, thereby consequently leading to moisture changes that will affect muscle fiber contraction and alter the texture of shrimp meat [44]. However, the color indicators exhibited a weaker correlation with the LF-NMR, which may be due to the fact that the color change is mainly caused by fat and pigmentation, and is weakly related with water splitting. In summary, the strong correlations between LF-NMR data and shrimp moisture content and texture properties indicate the potential of LF-NMR as a fast and nondestructive alternative method of detecting quality changes during shrimp drying.

### 3.5. Analysis of Modeling Results Based on Spectral Information

#### 3.5.1. Spectral Characteristics of Drying Processes

Figure 6 shows the average spectra of the ROIs in the shrimp samples. The spectral reflectance curves of the shrimp samples with different drying levels are smooth and exhibit the same trends across the entire wavelength region. As shown in Figure 6a, a prominent absorption peak is centered at approximately 480 nm, which is probably due to the presence of astaxanthin in the shrimp [45]. Astaxanthins present in the dermis of the carapace are bound to proteins, and when shrimp are heated at high temperatures, astaxanthin detaches from the proteins, causing red astaxanthin to become present. Another intense absorption peak occurred at approximately 960 nm, which was attributed to water absorption corresponding to the second overtone of O–H stretching [24]. Because water is the main component of shrimp, it absorbs the radiation of light waves and dominates the spectral characteristics between 950 and 1000 nm. Figure 6b shows the representative reflectance spectra of boiled and processed shrimp at different drying times (3, 6, 9, and 12 h). Over the wavelength region of 400–920 nm, the reflectance of boiled shrimp was greater than that in the dried samples, and the reflectance of dried shrimp decreased as the drying time increased. This phenomenon is related to moisture changes during shrimp-drying, especially to the mechanism of vapor diffusion [46]. Changes in muscle tissue and pigmentation during drying also contribute to this phenomenon.

#### 3.5.2. Prediction Models Using Whole Spectra

After spectral pretreatment, PLSR and LSSVM calibration models were established using the mean spectra from 400–1000 nm (224 bands) to predict quality changes in shrimp during drying. The main statistical parameters used to evaluate model performance are shown in Table 3. The two models exhibited reasonable and similar performance. For shrimp moisture content, both PLSR and LSSVM models yielded satisfactory results with *R*_p_ > 0.92 and RPD > 2.5. Both models performed well for the prediction set, with RPD values of 2.623 and 2.814, respectively, indicating that the LSSVM model is superior. For shrimp color (*L**, *a**, and *b**), the *R*_p_ values of *L**, *a**, and *b** obtained with the LSSVM model were 0.898, 0.919, and 0.906, respectively, showing excellent accuracy. Compared with the LSSVM results, the *R*_p_ values of *L**, *a**, and *b** obtained with the PLSR model were 0.853, 0.887, and 0.891, indicating a decrease of 0.045, 0.032, and 0.015, respectively. The performances of the PLSR and LSSVM models were much better than those obtained in a previous study by Wu et al. in which low *R*_p_ values of 0.864, 0.736, and 0.798 were achieved respectively for *L**, *a**, and *b** prediction in salmon [47]. Significant correlations between the color parameters (*L**, *a**, and *b**) and reflectance spectra could imply that the color changes indicate the shrimp chemical composition that indirectly influences the reflectance spectra. Compared to the PLSR model, the RMSEP for hardness and elasticity decreased from 32.663 N to 20.486 N and from 0.181 mm to 0.151 mm, respectively, in the LSSVM model, whereas RPD increased from 2.162 to 2.226 and from 2.118 to 2.208, respectively. These findings prove that the LSSVM model is more effective in terms of hardness and elasticity prediction, and demonstrate the potential of using HSI to estimate shrimp quality during the drying process.

#### 3.5.3. Prediction Models Using Characteristic Wavelengths

As multivariable (high-dimensional) data are extracted from hyperspectral images; they contain many inter-band correlations, resulting in long data processing times and low accuracy and robustness of the models [48,49]. After the SNV spectral pretreatment, the CARS algorithm was employed to identify the optimal wavelengths that carry the most information, which is useful for determining the moisture content, *L**, *a**, *b**, hardness, and elasticity. The number of Monte Carlo sampling runs was set to 1000, and the number of selected wavelengths was determined by 10-fold cross-validation. As a result, 42, 25, 39, 20, 29, and 18 optimal wavelengths were selected from the 400–1000 nm range, which occupied <19% of the entire wavelength range (224).

Based on the identified optimal wavelengths, simplified PLSR (CARS-PLSR) and LSSVM (CARS-LSSVM) models were established for the prediction of quality parameters of shrimp during the drying processes, and the results are presented in Figure 7. Compared with the PLSR and LSSVM models based on full spectra, the CARS-PLSR and CARS-LSSVM models achieved a better prediction result for all quality indicators (*L**, *a**, *b**, hardness, and elasticity) except moisture content, which could be attributed to the selection of effective wavebands during optimal wavelength selection in the CARS method. For shrimp moisture content, the RPD based on the characteristic wavelengths model was slightly lower than that determined using the full spectra because the process of filtering the characteristic wavelengths misses some important information. For shrimp color and texture, the prediction results of the characteristic wavelengths models were significantly improved, and the LSSVM models results were better than the PLSR model results. The RPD of the LSSVM model reached 2.541, 2.550, and 2.795 for *L**, hardness, and elasticity, respectively. Overall, it is reasonable to select the optimal wavelengths by employing the CARS method, which removed approximately 80% of the wavebands, significantly decreasing the data processing time and increasing the working efficiency. The newly developed model based on optimal wavelengths exhibits a powerful ability to predict the quality parameters of shrimp during drying.

### 3.6. Analysis of Modeling Results Based on Image Information

#### 3.6.1. Color Feature Information Extraction

The hyperspectral images at 647 nm, 550 nm, and 460 nm were used to synthesize *RGB* images as the target images for color feature extraction. The first- and second-order moment statistics for the *R*, *G*, and *B* components were calculated and listed in Table 4. Owing to the large amount of data, the color moment information of the eight samples was averaged. The first-order moment represents the average strength of the color component, whereas the second-order moment represents the color variance (i.e., non-uniformity) [29]. As shown in Table 4, the first-order moments show an overall increasing trend, and the second-order moments exhibit a decreasing trend; it indicates that the average intensity of the image color increases, and the color distribution becomes more uniform. These characteristics may be due to the oxidation of astaxanthin in shrimp with increasing drying time, resulting in a darker color. Because the *RGB* color space does not match human color perception, this space was converted into a visual-perception-oriented HSV space to calculate the histogram and quantify information. The mean grayscale values of the H, S, and V components are listed in Table 4. As the drying time increases, the overall S and V values increase, whereas the difference in H is small, indicating that shrimp images with different degrees of drying show less variation in hue.

#### 3.6.2. Texture Feature Information Extraction

As important as visual characteristics, texture features can also reflect differences in the chemical composition and structure of foods [50]. In this study, PCA was conducted for each individual image to evaluate the spatial variability of the samples; the top three principal component images (PC1, PC2, and PC3) with a cumulative contribution of 99.58% were selected for GLCM to obtain the contrast, correlation, energy, and homogeneity. The PCA process and average trends of the four textural features of the eight samples with different drying times are shown in Figure 8. It was clear that the contrast of the samples differed with increasing drying time, as denoted by the large differences in the gray value of the images, firstly exhibiting an increasing trend followed by a decrease in contrast (Figure 8a), which may be related to changes in the muscle texture during the shrimp-drying process. The correlation varies less (Figure 8b), fluctuating from 0.7 to 0.9, indicating that the texture uniformity of shrimp images with different drying levels is similar. As the drying time increases, the energy firstly decreases and then increases (Figure 8c). Homogeneity shows an opposite trend, reaching a minimum value at the seventh hour of drying (Figure 8d).

#### 3.6.3. Image Information Modeling Results

To verify whether the color and texture features of the hyperspectral images can be used to predict the quality indicators of shrimp during drying, nine color variables and four texture variables were selected and used to construct PLSR and LSSVM prediction models. The color variables were used to predict *L**, *a**, and *b**, the texture variables were used to predict hardness and elasticity, and 13 integration variables were used to predict moisture content. The PLSR and LSSVM model results based on image information are presented in Table 5. The LSSVM model yielded better predictions than the PLSR model. Specifically, the LSSVM model results for color were good, with RPD values of 1.642, 1.510, and 1.544 for *L**, *a**, and *b**, indicating that images can be used to predict shrimp color. However, the hardness and elasticity predictions were relatively poor, which may be because the amount of extracted textural information was not sufficient to accurately reflect shrimp hardness and elasticity. Overall, the models based on hyperspectral image information were inferior to those based on spectral data, which highlights the inadequacy of using only external image features to predict the quality indicators of shrimp during drying.

### 3.7. Analysis of Modeling Results Based on Fusion Mapping Feature Information

To further verify whether the integration of the image and spectral data from shrimp samples could optimize the prediction model and improve the accuracy for moisture content, color (*L**, *a** and *b**), and texture (hardness and elasticity), the variables from the optimal spectra and HSI color and texture information were integrated by feature-level fusion using the normalization technique. Thus, fusion data comprising the optimal wavelength of each indicator and 13 color and texture features were used to establish new PLSR and LSSVM models. The prediction results of full bands, characteristic bands, and fusion information are given and compared in Figure 9. Regarding shrimp moisture content (Figure 9a), the fusion models achieved limited improvement. The LSSVM model using full-band spectral information exhibited the best performance for dried shrimp (*R*_c_ = 0.959; *R*_p_ = 0.938; RPD = 2.814). For *L**, *a**, and *b** (Figure 9b–d), the fusion-based PLSR and LSSVM models exhibited substantial improvement. The LSSVM model was superior to the PLSR model, with RPD values for *L**, *a**, and *b** of 3.292, 2.753, and 3.211, indicating an increase in the prediction performance of 0.866, 0.172, and 0.859 than the PLSR model, respectively. For hardness and elasticity (Figure 9e,f), the fusion-based LSSVM model also showed excellent results compared to the fusion-based PLSR model, with the RPD values increasing from 2.612 to 2.807 and from 2.717 to 2.842, respectively. Thus, combining the internal components and external attributes of shrimp can more fully explain the color and texture changes of shrimp during drying, leading to better prediction results.

### 3.8. Visualization of Quality Indicators

A unique advantage of HSI technology compared with traditional spectroscopy or computer imaging technology is visualization of the prediction index of tested samples. Figure 10 visualizes the moisture content, *L**, *a**, *b**, hardness, and elasticity of shrimp generated by the optimal model selected from the modeling results. In the maps, the distribution of shrimp moisture content is expressed by a linear color bar ranging from blue (low value) to red (high value). The boiled shrimp have a high moisture content of 73.02%. The moisture content of the samples then gradually decreases with drying time to a final value of 35.02%. As for shrimp color, *L**, *a**, and *b** values tend to decrease during the drying process. Although this difference cannot be observed by visual inspection, the spatial distribution of color features within the shrimp was detected in the final distribution map generated by analyzing the hyperspectral image of the sample. Furthermore, the visualization maps show a clear increase in the hardness of shrimp, whereas the distribution of elasticity is more complex. Thus, the distribution maps of shrimp moisture content, color, and texture provide an intuitive analysis of changes in the quality reference values for dried shrimp, which are unlikely to be observed by the naked eye or an *RGB* image.

In more detail, the moisture content distribution is non-uniform and asymmetric. This may be attributed to complex changes in protein decomposition, lipid oxidation, etc. Furthermore, drying temperature and time may accelerate the degradation of ruptured tissue and cells in meat, leading to further uneven water loss [51]. Following shrimp drying, the *L** value decreases with the oxidization of myoglobin and hemoglobin into metmyoglobin and methemoglobin [52]. In addition, the color of shrimp becomes orange, with yellow or orange-red colors resulting from the oxidization reaction and the presence of astaxanthin. The hardness and elasticity of shrimp exhibits a non-uniform distribution that is related to the distribution of fat, pigments, and collagen [53]. In summary, HSI combined with data fusion can achieve the nondestructive detection and visualization of shrimp color and texture during drying. Specifically, the distribution maps of quality indicators generated using HSI clarify the location and movement of water, color, and textures through the shrimp samples during the hot air drying process. Such maps help consumers intuitively understand the dynamic changes in shrimp quality and the shelf life of dried shrimp production. Thus, we present a valid alternative to traditional methods of monitoring shrimp drying that has substantial potential for further development and can be applied to detect freshness or other indexes during aquatic production.

## 4. Conclusions

In this study, we described the changes in shrimp quality, evaluated the correlation between shrimp water distribution state and other quality indices, and combined spectral and image information of the hypercube to monitor shrimp quality changes during the drying process. Throughout the process, the moisture content showed a downward trend, the hardness and elasticity reached 344.78 N and 1.56 mm, respectively, and the color turned bright yellow at the end of drying. Significant correlations between the moisture content, TPA parameters (hardness, adhesiveness, elasticity, stickiness, and chewiness), and LF-NMR parameters (T_21_, T_22_, and T_23_) were observed. The HSI system in the spectral range of 400–1000 nm was used to monitor the quality changes (moisture content, *L**, *a**, *b**, hardness, and elasticity) of shrimps during drying. The results demonstrate the following: first, the ability of the HSI method to evaluate the quality changes of shrimps during drying; second, the optional wavelengths selected by the CARS method carried the most effective information, which reduced the spectral dimension and accelerated the calibration process; and finally, the spectral information model predicts better than the image color and texture information model, and the LSSVM built by combining image information with spectral information in characteristic bands has powerful and accurate prediction capabilities. Thus, HSI can be utilized to visualize the quality changes in shrimps in a pixel-wise manner both quantitatively and automatically, reducing the overall production cost, saving time, and avoiding subjectivity and discrepancies.

## Figures and Tables

**Figure 1 foods-11-03179-f001:**
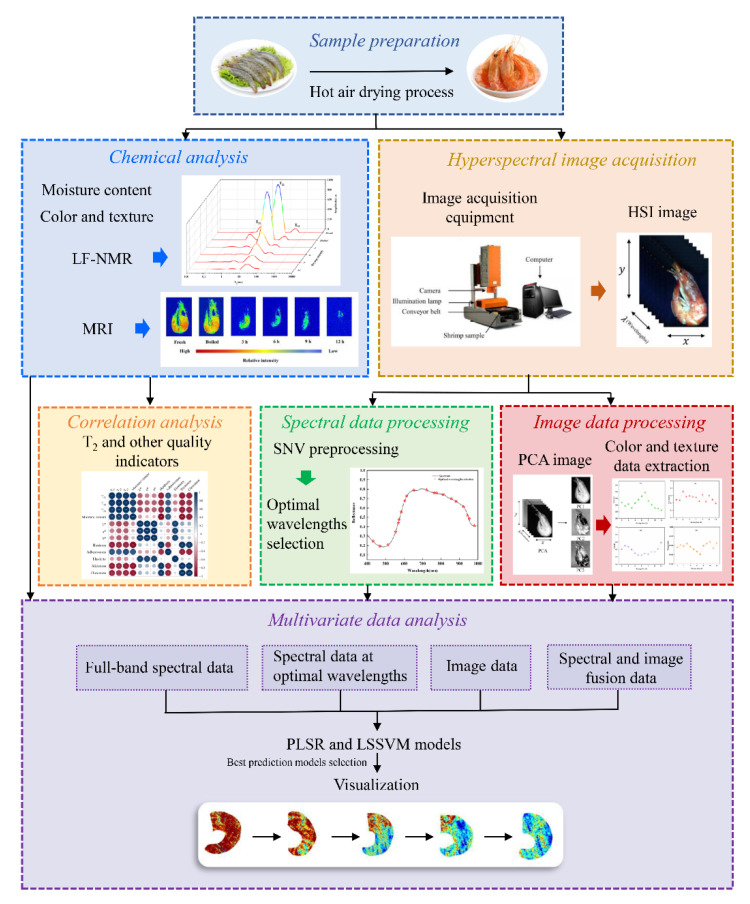
Flowchart of data analyses.

**Figure 2 foods-11-03179-f002:**
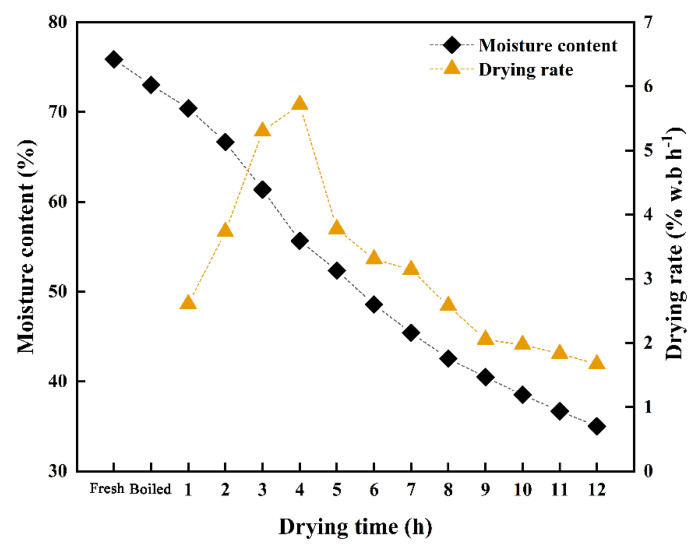
Averaged measured moisture contents of all samples at different drying levels.

**Figure 3 foods-11-03179-f003:**
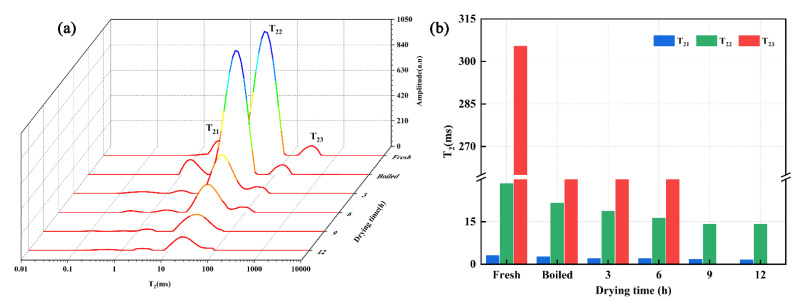
(**a**) Distribution of T_2_ relaxation spectra and (**b**) change of T_2_ relaxation times obtained by multi-exponential fitting of the continuously distributed Carr–Purcell-Meiboom-Gill relaxation curve of different shrimp samples during drying.

**Figure 4 foods-11-03179-f004:**
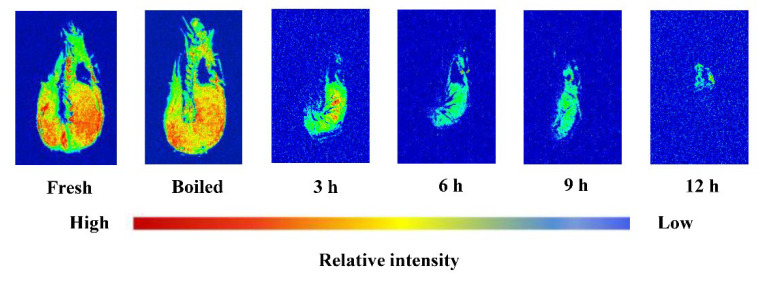
T_2_ weighted MRI images of shrimp dried by hot air drying at different levels.

**Figure 5 foods-11-03179-f005:**
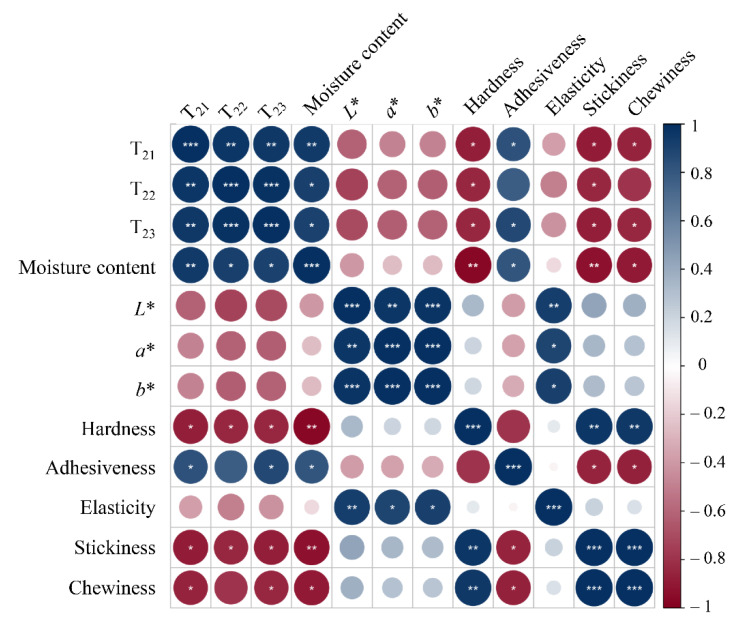
Correlation analysis between LF-NMR relaxation parameters and moisture content, color, texture of shrimp during drying.

**Figure 6 foods-11-03179-f006:**
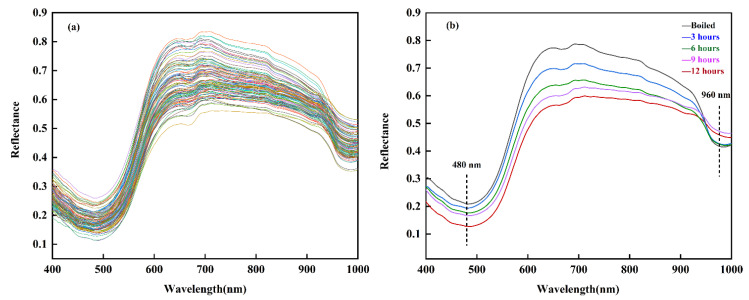
(**a**) Mean reflectance spectra of the ROIs in shrimp samples with different drying levels and (**b**) reflectance spectra at different drying times (boiled, 3, 6, 9, and 12 h).

**Figure 7 foods-11-03179-f007:**
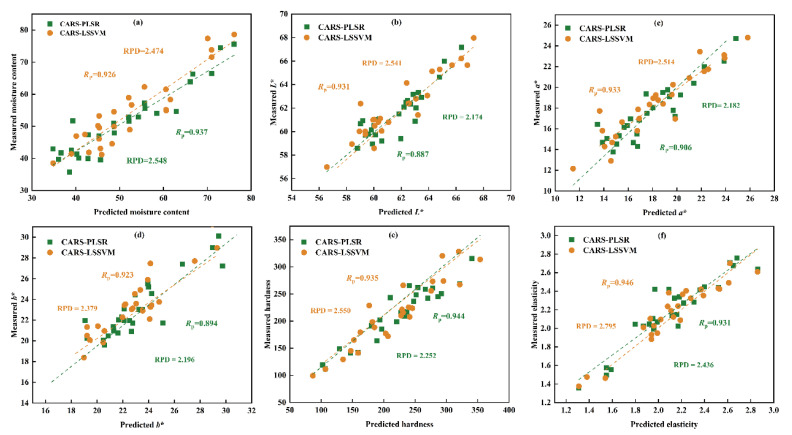
Comparison of CARS-PLSR and CARS-LSSVM in terms of (**a**) moisture content, (**b**) *L**, (**c**) *a**, (**d**) *b**, (**e**) hardness, and (**f**) elasticity based on quantitative analysis models in shrimp during drying.

**Figure 8 foods-11-03179-f008:**
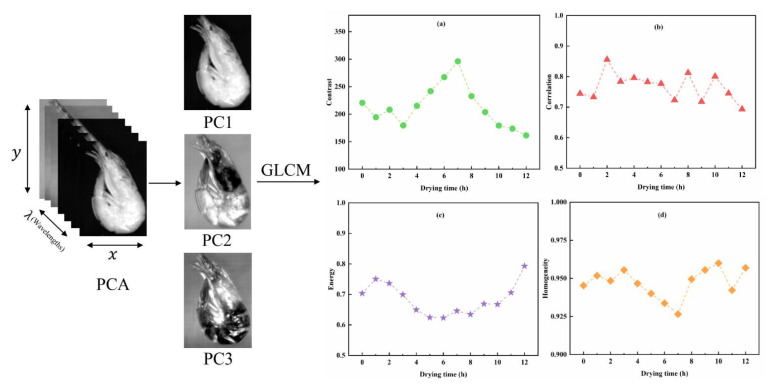
PCA process and texture features of shrimp samples, (**a**–**d**) stand for the change of contrast, correlation, energy, and homogeneity, respectively.

**Figure 9 foods-11-03179-f009:**
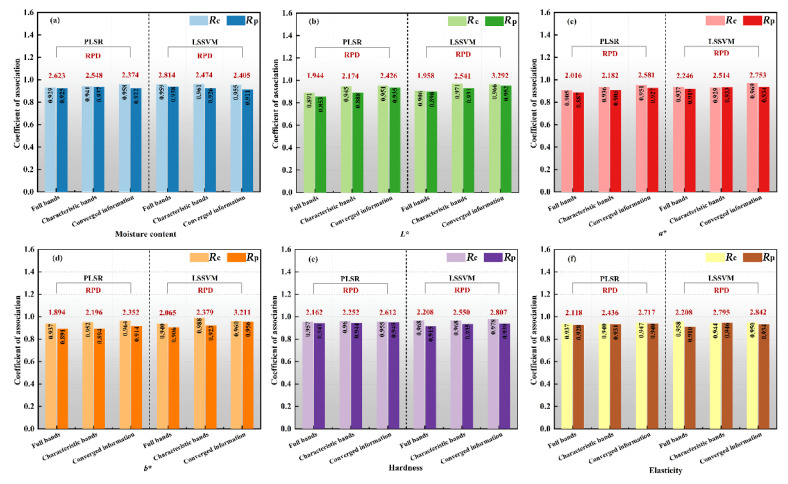
Comparison results of (**a**) moisture content, (**b**) *L**, (**c**) *a**, (**d**) *b**, (**e**) hardness, and (**f**) elasticity models.

**Figure 10 foods-11-03179-f010:**
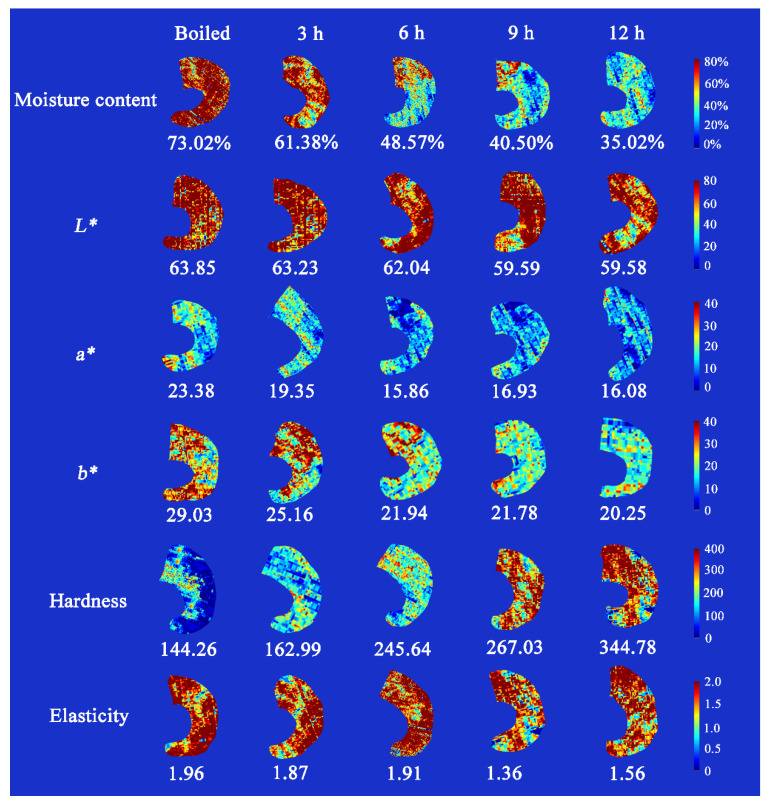
Moisture content, color (*L**, *a**, and *b**) and texture (hardness, and elasticity) visualization map of shrimp at different drying times (boiled, 3, 6, 9 and 12 h).

**Table 1 foods-11-03179-t001:** Effects of hot air drying on the texture of shrimp.

Drying Time (h)	Hardness (N)	Adhesiveness (N)	Elasticity (mm)	Stickiness (mJ)	Chewiness (mJ)
Fresh	127.96 ± 15.44 ^f^	1.90 ± 0.31 ^a^	0.83 ± 0.13 ^f^	2.84 ± 0.58 ^h^	2.38 ± 0.74 ^h^
Boiled	144.26 ± 16.24 ^f^	1.48 ± 0.17 ^c^	1.96 ± 0.23 ^c,d,e^	12.23 ± 1.99 ^g^	24.37 ± 6.85 ^g^
1	159.64 ± 28.24 ^e,f^	1.45 ± 0.18 ^c^	1.90 ± 0.22 ^d,e^	14.61 ± 3.32 ^f,g^	28.44 ± 9.41 ^f,g^
2	168.89 ± 34.57 ^e,f^	1.36 ± 0.14 ^c^	1.99 ± 0.21 ^c,d,e^	17.06 ± 3.96 ^e,f,g^	34.63 ± 11.19 ^f,g^
3	162.99 ± 20.22 ^e,f^	1.14 ± 0.11 ^d^	1.87 ± 0.16 ^d,e^	14.55 ± 1.24 ^f,g^	27.21 ± 4.07 ^f,g^
4	197.26 ± 45.33 ^d,e^	1.48 ± 0.19 ^c^	1.74 ± 0.18 ^e^	19.40 ± 4.70 ^d,e,f^	34.44 ± 11.17 ^f,g^
5	231.38 ± 41.98 ^c,d^	1.37 ± 0.11 ^c^	1.89 ± 0.18 ^d,e^	20.74 ± 4.40 ^c,d,e,f^	39.41 ± 10.75 ^e,f,g^
6	245.64 ± 52.16 ^c^	1.67 ± 0.32 ^b^	1.91 ± 0.31 ^d,e^	21.20 ± 6.82 ^c,d,e,f^	41.67 ± 18.05 ^d,e,f,g^
7	233.56 ± 56.55 ^c,d^	0.25 ± 0.06 ^f,g^	2.09 ± 0.20 ^c,d^	22.15 ± 5.17 ^c,d,e^	46.59 ± 12.44 ^c,d,e,f^
8	265.54 ± 39.32 ^b,c^	1.00 ± 0.12 ^d^	2.05 ± 0.40 ^c,d^	26.55 ± 5.60 ^c^	55.78 ± 21.14 ^c,d,e^
9	267.03 ± 58.75 ^b,c^	0.33 ± 0.06 ^e,f,g^	1.36 ± 0.23 ^a,b^	26.61 ± 6.87 ^c^	63.47 ± 19.29 ^c^
10	252.94 ± 44.79 ^b,c^	0.41 ± 0.10 ^e,f^	1.23 ± 0.34 ^b,c^	25.95 ± 8.07 ^c,d^	60.13 ± 25.82 ^c,d^
11	296.95 ± 63.08 ^b^	0.47 ± 0.08 ^e^	1.51 ± 0.26 ^a^	37.14 ± 10.70 ^b^	92.36 ± 24.28 ^b^
12	344.78 ± 44.22 ^a^	0.20 ± 0.02 ^g^	1.56 ± 0.35 ^a^	45.01 ± 9.78 ^a^	116.50 ± 34.33 ^a^

Note: All data are presented as mean ± standard error. Mean values with different letters within each line are significantly different (*p* < 0.05) with respect to processing.

**Table 2 foods-11-03179-t002:** Effects of hot air drying on the color of shrimp.

Drying Time (h)	*L**	*a**	*b**	Δ*E*
Fresh	40.71 ± 1.22 ^f^	1.22 ± 0.55 ^f^	3.08 ± 1.39 ^h^	-
Boiled	63.85 ± 1.22 ^a^	23.38 ± 2.30 ^a^	29.03 ± 2.46 ^a^	41.304 ± 2.34 ^a^
1	63.89 ± 1.13 ^a^	21.33 ± 1.79 ^b^	25.85 ± 1.96 ^b^	38.261 ± 1.97 ^b^
2	63.68 ± 1.17 ^a^	20.02 ± 1.39 ^b,c^	24.19 ± 2.08 ^b,c,d^	36.460 ± 2.12 ^b,c^
3	63.23 ± 2.17 ^a,b^	19.35 ± 1.87 ^c^	25.16 ± 1.84 ^b,c^	36.493 ± 1.40 ^b,c^
4	63.12 ± 1.61 ^a,b,c^	20.24 ± 1.57 ^b,c^	24.09 ± 1.65 ^b,c,d^	36.158 ± 2.26 ^c^
5	61.79 ± 0.88 ^b,c^	17.56 ± 1.61 ^d^	24.15 ± 1.68 ^b,c,d^	34.043 ± 1.41 ^d^
6	62.04 ± 1.52 ^b,c^	15.86 ± 2.74 ^d,e^	21.94 ± 1.37 ^e,f,g^	32.144 ± 1.49 ^e,f^
7	61.62 ± 2.24 ^c,d^	16.60 ± 2.40 ^d,e^	23.35 ± 1.79 ^c,d,e^	33.025 ± 2.62 ^d,e^
8	60.23 ± 1.28 ^d,e^	15.99 ± 1.95 ^d,e^	22.25 ± 1.77 ^d,e,f^	31.152 ± 2.09 ^e,f,g^
9	59.59 ± 1.58 ^e^	16.93 ± 1.28 ^d,e^	21.78 ± 1.79 ^e,f,g^	30.938 ± 1.52 ^f,g^
10	59.63 ± 1.68 ^e^	15.56 ± 1.06 ^d,e^	20.36 ± 1.75 ^f,g^	29.433 ± 1.60 ^g^
11	59.93 ± 0.56 ^e^	15.44 ± 1.73 ^e^	19.83 ± 2.56 ^g^	29.289 ± 1.80 ^g^
12	59.58 ± 0.79 ^a^	16.08 ± 1.31 ^d,e^	20.25 ± 2.42 ^f,g^	29.617 ± 1.28 ^g^

Note: All data are presented as mean ± standard error. Mean values with different letters within each line are significantly different (*p* < 0.05) with respect to processing. “-”represents the blank.

**Table 3 foods-11-03179-t003:** Prediction models for moisture content, *L**, *a**, *b**, hardness, and elasticity values using 224 wavelengths.

Parameters	Pre-Processing	Model	Calibration Set		Prediction Set		RPD
			*R* _c_	RMSEC	*R* _p_	RMSEP	
Moisture content	SNV	PLSR	0.929	4.369	0.925	4.512	2.623
	SNV	LSSVM	0.959	3.378	0.938	4.312	2.814
*L**	SNV	PLSR	0.891	0.975	0.853	1.257	1.944
	SNV	LSSVM	0.906	1.002	0.898	1.031	1.958
*a**	SNV	PLSR	0.905	1.010	0.887	1.249	2.016
	SNV	LSSVM	0.937	0.998	0.919	1.181	2.246
*b**	SNV	PLSR	0.937	1.045	0.891	1.325	1.894
	SNV	LSSVM	0.940	0.875	0.906	0.945	2.065
Hardness	SNV	PLSR	0.957	16.545	0.941	32.663	2.162
	SNV	LSSVM	0.968	12.758	0.915	20.486	2.226
Elasticity	SNV	PLSR	0.937	0.116	0.928	0.181	2.118
	SNV	LSSVM	0.958	0.073	0.910	0.151	2.208

**Table 4 foods-11-03179-t004:** Extracted image feature information of color.

Drying Times (h)	First Order Moments			Second Order Moments			*H*	*S*	*V*
	*R*	*G*	*B*	*R*	*G*	*B*			
Boiled	62.284	56.703	62.217	38.837	35.045	27.693	0.487	0.595	0.307
1	61.977	55.456	60.337	37.209	34.035	27.374	0.482	0.605	0.301
2	60.761	57.174	64.857	36.829	33.947	26.746	0.496	0.605	0.311
3	61.490	55.945	62.807	38.330	34.523	27.186	0.495	0.615	0.309
4	60.892	55.974	62.138	37.353	33.333	25.926	0.488	0.615	0.308
5	61.110	56.265	62.507	35.692	32.388	25.120	0.490	0.607	0.309
6	63.739	58.015	62.952	34.702	31.067	23.947	0.479	0.613	0.318
7	65.538	59.785	62.133	35.201	30.905	24.029	0.459	0.609	0.322
8	60.655	55.422	60.712	36.566	31.897	24.553	0.481	0.623	0.308
9	64.497	57.990	61.884	36.441	31.830	24.614	0.479	0.600	0.317
10	63.540	58.484	63.141	38.164	32.466	24.667	0.476	0.622	0.322
11	63.438	57.540	62.576	36.181	31.338	23.793	0.483	0.621	0.321
12	68.387	61.024	64.176	36.032	31.173	23.693	0.471	0.609	0.334

**Table 5 foods-11-03179-t005:** Results of PLSR and LSSVM models based on image information.

Parameters	PLSR			LSSVM		
	*R* _p_	RMSEP	RPD	*R* _p_	RMSEP	RPD
Moisture content	0.695	9.569	1.065	0.730	8.564	1.197
*L**	0.690	2.323	1.243	0.798	1.845	1.642
*a**	0.701	2.570	1.317	0.762	1.901	1.510
*b**	0.655	2.609	1.287	0.794	2.010	1.544
Hardness	0.591	46.198	1.088	0.685	40.103	1.395
Elasticity	0.581	0.332	1.192	0.698	0.207	1.404

## Data Availability

The data presented in this study are available on request from the corresponding author.

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
