# Peer review of "Real-Time Monitoring of the Quality Changes in Shrimp (Penaeus vannamei) with Hyperspectral Imaging Technology during Hot Air Drying"

_foods, 2022, doi:10.3390/foods11203179_

Round 1

Reviewer 1 Report

The reviewed manuscript concerns the study of correlation between water distribution state in shrimps and quality indices like color parameters, texture parameters. I highly appreciate the scientific level of the article, especially the statistical methods used. The combined spectral and image information of the hypercube to monitor shrimp quality changes during the drying process also seems very interesting. The paper has a good layout  with good organised “Introduction” section and and a very well presented graphic interpretation of the obtained results. 

Author Response

Point : The reviewed manuscript concerns the study of correlation between water distribution state in shrimps and quality indices like color parameters, texture parameters. I highly appreciate the scientific level of the article, especially the statistical methods used. The combined spectral and image information of the hypercube to monitor shrimp quality changes during the drying process also seems very interesting. The paper has a good layout with good organized “Introduction” section and a very well presented graphic interpretation of the obtained results.

Response : Thank you very much for your comments. Thank you very much for your affirmation of this paper. Your high praise for this paper has given me sufficient confidence to continue my efforts in the next research work. Thanks again!

Reviewer 2 Report

My comments

The manuscript entitled “Real-time monitoring of the quality changes in shrimp (Penaeus vannamei) with hyperspectral imaging technology during hot air drying" by Xu, et al. described a hyperspectral imaging (HIS) approach for monitoring water content of drying shrimps. The manuscript is mostly well-written after a minor revision base on my comments below.

Major concerns

1.     The manuscript is very long and limited number of references have been give. For instance, only 19 references were cited in the introduction section. For a paper of this length I think the references should be about two times the current number.

2.     The authors did not mention the drying rate, perhaps this is not important in their experiments. Regardless, the authors should say something about the drying rate and its significance to the work.

Minor comments

3.     Line 14: sate?

4.     Line 328: “…at different drying stages of was measured…” Something missing here.

5.     Lines 367-369: “….the correlation between LF-NMR 367 results and the physicochemical parameters of shrimp during the drying process remains 368 deeper exploration.” The authors should revise this as the message isn’t clear here.

6.     Line 500: change “…a decreasing in contrast (Figure 8a), which may be related with changes in the” to “… a decrease in contrast (Figure 8a), which may be related to changes in the”.

Author Response

Point 1 :The manuscript is very long and limited number of references have been give. For instance, only 19 references were cited in the introduction section. For a paper of this length I think the references should be about two times the current number.
Response 1: We are grateful for the suggestion. As suggested by the reviewer, we have added references to the introduction and other parts of the article. The references cited have been marked in the "References" section. Thanks again!
Point 2 :The authors did not mention the drying rate, perhaps this is not important in their experiments. Regardless, the authors should say something about the drying rate and its significance to the work.
Response 2: Thank you very much for your kind advice. As suggested by the reviewer, we have added the analysis of drying rate in the section of "3.1.1 Moisture content analysis" and modified Figure 2, as detailed in Lines 251-257, Line 269 and Figure 2 of the revised manuscript. Thanks again!
Point 3 :Line 14: sate?
Response 3: We are grateful for the question. We apologize for the confusion caused by the lack of clarity here. We have made the changes you requested in Line 14 in the revised manuscript. We change “sate” to “distribution”.
Point 4 :Line 328: “…at different drying stages of was measured…” Something missing here.
Response 4: We are grateful for the suggestion. We have made the changes you requested in Line 329-331 in the revised manuscript. Thanks again!
Point 5 :Lines 367-369: “….the correlation between LF-NMR 367 results and the physicochemical parameters of shrimp during the drying process remains 368 deeper exploration.” The authors should revise this as the message isn’t clear here.
Response 5: We are grateful for the suggestion. As suggested by the reviewer, we have revised Line 367-369, please see Line 370-372 in the revised manuscript.
Point 6 :Line 500: change “…a decreasing in contrast (Figure 8a), which may be related with changes in the” to “… a decrease in contrast (Figure 8a), which may be related to changes in the”.
Response 6: Thank you very much for your kind advice. As suggested by the reviewer, we have made the changes in Line 502-503 in the revised manuscript. We changed “…a decreasing in contrast (Figure 8a), which may be related with changes in the” to “… a decrease in contrast (Figure 8a), which may be related to changes in the”.
